# Moving Beyond the Checklist—An Enhanced Approach to Context-Driven Overuse Injury Prevention in Overhead Youth Athletes

**DOI:** 10.3390/jcm14030971

**Published:** 2025-02-03

**Authors:** Rachel Lau, Swarup Mukherjee

**Affiliations:** Physical Education and Sports Science Department, National Institute of Education, Nanyang Technological University, Singapore 637616, Singapore; nie20.lytr@e.ntu.edu.sg

**Keywords:** context-driven, injury prevention, overuse injury, overhead youth athlete

## Abstract

The increasing discussions regarding the research-to-practice gap in sport injury prevention have led to a growing focus on the significance of context in injury prevention programs. A context-driven injury prevention program is advantageous to address and enhance injury prevention efforts in specific populations. Considering the prevalent concern for overuse injuries among overhead youth athletes, and the developmental and contextual factors unique to this population, this review introduces the OverHead Youth Athlete (OH-YA) overuse injury prevention process. Tailored to address the challenges of working with overhead youth athletes, the context-specific four-step framework includes the following: (1) development of a context-specific instrument with strong sample representation, (2) determination of the context-specific injury burden magnitude and associated factors, (3) development of a context-specific intervention with end-users, and (4) evaluation of the context-specific intervention. This framework would likely help maximize the use of typically limited resources in youth sports. Using shoulder overuse injuries among overhead youth athletes as a case study, the OH-YA overuse injury prevention process provides an enhanced approach to context-driven overuse injury prevention while illustrating the importance and integration of context to minimize the “research-to-practice” gap.

## 1. Introduction

The significance of context in sport injury prevention has recently received growing attention in parallel with focused efforts to address the challenge of translating research findings into real-world conditions, otherwise known as the research-to-practice gap [1,2]. With progress in research in the field of sports injury prevention, it has become apparent that consideration of the intended end-users (i.e., implementation context) is imperative prior to intervention dissemination to improve the effectiveness of preventive measures in real-world scenarios [3,4]. This has led to the development of the “context-driven sequence of prevention” model by Bolling et al. [1], which builds on the original “sequence of prevention” model by van Mechelen et al. [5]. The four steps of the “context-driven sequence of prevention” are (1) injury and its context, (2) contextual determinants, (3) context-driven intervention, and (4) evaluation of multiple measures [1]. Essentially, the model calls for an all-encompassing focus on the intended context to drive the injury prevention process, with the goal of enhancing the application of injury prevention research in real-world settings.

By tailoring interventions to fit specific contexts, it becomes possible to integrate these interventions directly into the athlete’s everyday training and conditioning environment to enable immediate application and evaluation of effectiveness [4]. It was further suggested in a recent systematic review and meta-analysis among female football players that a tailored approach may facilitate intervention buy-in and adherence from end-users, consequently improving program effectiveness [6]. Moreover, this strategy for a context-specific intervention may be more efficacious than the traditional approach of adopting controlled, ideal-condition interventions for real-world use, which often demands additional resources and may not achieve anticipated outcomes under less controlled conditions [1]. This is especially concerning in situations where the intervention has to be modified to suit the context appropriately (e.g., limited equipment, limited time) which may affect the program’s efficacy [7]. For instance, it was documented in a qualitative study among junior high school teachers and students that the implemented injury prevention program (i.e., iSPRINT) was not time-efficient as participants had to spend five to eight minutes setting up the necessary equipment, and a similar amount of time to keep them afterwards [8]. This led them to make their own modifications to the program (i.e., execute the balance-related exercises without a wobble board), which may have affected the intended effects of the program [8]. Therefore, a context-driven approach that considers the intended and representative sport population would be advantageous and can be reasonably positioned as a preferred approach to effectively address sports injuries and enhance injury prevention efforts.

Considering the documented high shoulder overuse injury prevalence (9–49%) and long-term detrimental effects (e.g., growth disturbances, reduced future quality of life) of overuse injuries, the unique population of overhead youth athletes is one that would benefit from a specifically tailored injury prevention process [9,10]. Despite their physical, psychological, and social benefits, sports participation without adequate rest and recovery has been demonstrated to lead to an accumulation of microtrauma and consequently overuse injuries [11,12]. This is particularly concerning among overhead youth athletes as overhead sports are characterized by repetitive forceful use of the arm. To illustrate, in performing the fastball pitch, the shoulder of youth baseball pitchers demonstrated a maximum internal rotation velocity of 6950 ± 1520 °/s during arm acceleration and a proximal force of 391 ± 82 N near ball release [13]. Furthermore, considering the developing skeletal and muscular systems of these growing individuals, such repetitive immense forces on their immature shoulders greatly increases their risk for shoulder overuse injuries.

The magnitude burden of overuse injuries among overhead youth athletes has remained consistent despite ongoing injury prevention efforts [14,15]. For instance, within the overhead sport of volleyball, a prevalence study conducted in 2023 among Singapore’s youth volleyball athletes found a shoulder overuse injury prevalence of 28.7% [16], which is comparable to the shoulder overuse injury prevalence of 27.8% reported among high school girls’ volleyball in 2015 [17]. In a different study conducted among high school female volleyball athletes in 2017 in the United States, the prevalence rate for shoulder pain that was unrelated to a traumatic injury was 40% [18]. These consistently high numbers suggest that youth sport is a unique context where a specific approach to the injury prevention process may be more effective. Unlike adult athletes, youth athletes are still physically and cognitively developing and maturing, which further supports the notion that a more contextually specific approach to injury prevention may be warranted to address the uniqueness of this population [19,20]. Despite these numerous concerns, seemingly no injury prevention process framework currently exists to address shoulder overuse inures within the specific context of overhead youth athletes. Therefore, we propose a specific approach for overuse injury prevention in overhead youth athletes, while explicitly integrating and emphasizing the importance of context within the injury prevention process. An enhanced, sequenced and inclusive model for overuse injury prevention in overhead youth athletes, known as the OverHead Youth Athlete (OH-YA) overuse injury prevention process is presented below (Figure 1), while utilizing shoulder overuse injuries among competitive overhead youth athletes as a case study. The OH-YA overuse injury prevention process builds on the “context-driven sequence of prevention” by introducing a preliminary step of instrument validation to enhance data accuracy. The framework will also likely help maximize the use of limited resources in youth sports.

## 2. Step 1: Ensure That the Instrument Used to Determine Injury Magnitude Burden Is Valid for the Context

The importance of context specificity begins even before the injury problem is investigated—it begins with the instrument used to determine the injury problem. A validated instrument is necessary to ensure an accurate measure of the injury burden magnitude. As represented by the first step of the OH-YA overuse injury prevention process, the instrument to be used needs to be developed specifically for the context, which can be achieved through adequate validation among a strongly representative sample of the population of interest (i.e., intended context) (Figure 2). This extends to the processes involved during instrument development, which should occur prior to instrument dissemination for validation. These processes should include seeking feedback from intended users on the instrument, such as the suitability of the language, sentence phrasing, and instrument length to enhance the context specificity and end-user acceptability of the instrument. Once this is ensured, the instrument can be deemed to be in a state of readiness to be disseminated for validation.

In our case, to determine the magnitude of overuse injuries among youth athletes, we developed a specific overuse injury questionnaire known as the Youth Overuse Injury Questionnaire (YOvIQ) [21]. YOvIQ was developed by building on the established updated Oslo Sport Trauma Research Center Overuse Injury Questionnaire (OSTRC-O2) to make it context-specific to youth athletes—this included adjusting the recall period, reorganizing the question order, emphasizing key words by bolding and underlining them, and adding a short, open-ended question for injury description [21,22].

Prior to dissemination for further validation, two content experts (in sports injury epidemiology and in sports science and medicine) and seven end-users (youth volleyball athletes) provided feedback [21]. The initial draft of YOvIQ was shared with the context experts who provided feedback via email. Subsequently, an online feedback session was conducted with the youth athlete end-users on the second draft of YOvIQ, to enhance the suitability of the questionnaire. Notably, it should be highlighted that when seeking feedback from youth athletes, it is important to emphasize the anonymity of their feedback, to help create a safe environment where they feel comfortable to share their opinions freely [21]. The inputs from both content experts and end-users were immensely valuable to prompt appropriate changes to the questionnaire, thus purposefully contributing to the content validity. As the questionnaire was developed specifically for use among competitive youth athletes, it was subsequently disseminated together with the OSTRC-O2 to competitive youth athletes to evaluate convergent validity [22].

In line with being context-specific, ensuring strong sample representation is essential during instrument validation. A total of 227 competitive youth athletes from 14 different sports were involved in establishing the validity of YOvIQ [21]. This was necessary as the developing cognitive abilities and limited injury experience of youth athletes may influence their ability to independently comprehend an injury questionnaire designed for adult athletes, limiting the accuracy of the data [19]. Moreover, apart from contributing to the development of a context-specific instrument, a diverse sample within the population of interest would enhance information accuracy and value of findings [23]. This primary step of validating an instrument prior to its use in the intended population of interest is critical as it lays the foundation for the overall context-specific injury prevention process and contributes to ensuring an accurate representation of the injury problem.

By specifically tailoring the questionnaire to make it suitable for use among competitive youth athletes, seeking expert and end-user feedback, and further validating its use among youth athletes from a diverse range of sports, YOvIQ can be regarded as a highly validated instrument within the specific context of competitive youth athletes. This comprehensive validation process of YOvIQ also offers enhanced accuracy of the data collected, enabling better tailored context-specific preventive interventions for increased effectiveness. 

## 3. Step 2: Determine the Injury Burden Magnitude and Associated Factors Within the Context

With a context-specific instrument, a valid and accurate representation of the injury burden magnitude within the population of interest can be ensured. Depending on the resources available in the context (e.g., participant interest, funding availability), the injury burden magnitude and associated factors may be determined either retrospectively or prospectively. Importantly, if a retrospective study design is used, it would be desirable to restrict the recall period to a maximum of 12 months to minimize recall bias [24]. If a prospective study design is utilized, researchers should be cognizant of the possibilities of non-response and attrition, especially among youth athletes due to loss of motivation or the study being too time-consuming [25,26].

Regardless of study design utilized, definitions are necessary, such as the population of interest (e.g., age and level of sport participation), the injury (e.g., type of symptoms), and the type of sport (e.g., a specific sport, a category of sports, or all sports). Establishing these definitions is necessary to ensure data accuracy in addressing the objective(s) of the study and for future comparison across studies. The valid data collected using the context-specific instrument may then be critically analyzed to categorize injuries based on their burden magnitude (i.e., injury prevalence and injury severity) into severe, moderate, and mild injury categories. Having these categorizations is essential as it provides a quick overview of injury levels within the context of interest (e.g., a sports team), which can help prioritize prevalent or pressing injury concerns to allow for more efficient allocation of limited resources. Additionally, the strongly representative and valid data collected may also be further analyzed to identify factors associated with the injury within the context (e.g., age, level of sport participation, gender), which will assist coaches and/or athletic trainers to promptly interpret and refine their injury management approaches.

To illustrate, we determined the prevalence of shoulder and elbow overuse injuries among 434 competitive overhead youth athletes in Singapore with a 3-month retrospective study design [16]. Importantly, definitions for the study were established by the inclusion criteria, which were as follows: (1) age range between 12 and 18 years of age, (2) the primary sport participated in is an overhead sport where the arm is repetitively raised above the head to strike or throw an object, and (3) was training with an aim to compete in at least one tournament in the upcoming season. The presence of an overuse injury was registered if participants reported anything less than full participation ability in training or competitions. Notably, in determining the shoulder and elbow overuse injury prevalence, it was demonstrated that shoulder overuse injuries (31.3%) were a more pressing concern than elbow overuse injuries (9.2%). Further investigation into the associated factors suggested that older participants (15 to 18 years of age) were at increased odds of shoulder overuse injuries as compared to younger participants 12 to 14 years of age (OR, 1.65; 95% CI, 1.10–2.49). Similarly, participants training more than 11 h per week were at increased odds of shoulder overuse injuries (OR, 2.64; 95% CI, 1.31–5.30) as compared to those training one to five hours per week. Considering the relatively large sample size of 434 participants across 11 overhead sports, the findings from this study, and similar studies alike, can provide valuable insights into injury patterns and risk factors for athletes in these sports. These insights may also contribute to the development of more targeted injury prevention interventions to foster safer sport participation while enhancing performance.

Having a clear overview of the injury problem within the context is vital to allow for the strategic allocation of resources to address the most pressing injury concern, especially among youth sport teams where resources are limited [27,28]. The processes of Step 2 are illustrated in Figure 3, as the second step of the OH-YA overuse injury prevention program.

## 4. Step 3: Include the Context During Injury Prevention Program Development

With an understanding of the injury burden magnitude through the use of a validated instrument, the corresponding injury prevention program can subsequently be developed, which includes a clear representation of injury priority and important risk factors. Considering the importance of ecological validity, significant emphasis must be placed at this step of the development process of an injury prevention program, where an understanding of the implementation context is essential (Figure 4).

Consideration for the implementation context during program development is critical as it is likely to improve program acceptance and uptake, maximize the use of limited resources, and consequently reduce the frequently discussed “research-to-practice” gap during real-world program implementation [1,4,29,30]. This can be achieved through collaboration with a strongly representative sample of the context during program development [3,31,32]. This should include experts who are acutely aware of the culture, practices, and even implementation issues of the specific context. In the context of youth sport injury prevention, it would be advisable for the representative sample to include local context experts (e.g., youth sport coaches, Physical Education teachers), practitioners (e.g., sports physicians, pediatricians, physiotherapists), end-users (e.g., youth athletes), and content experts (e.g., researchers in sports science and sports injury epidemiology), who will be able to provide feedback on what is considered practical to be readily accepted and easily implemented into the context.

The importance of seeking feedback from experts was demonstrated in our development of the Singapore Youth Shoulder Overuse Injury Prevention Programme (YoSO-IPP), where we consulted local experts via a two-round Delphi technique [33]. In our study, as part of the iterative multistage process, the experts utilized a five-point Likert Scale to provide feedback and were encouraged to leave comments for anonymous sharing with the other experts in the next round. To illustrate an example, some experts left comments in the first round of the Delphi technique regarding the potential difficulty of implementing the cat–cow exercise among the targeted participants of 12–18-year-old athletes, due to its technical difficulty. This led the research team to modify the cat–cow exercise to make it easier to perform, which was included in the second round and subsequently reached consensus (i.e., 75% agreement) among the experts [33]. If feedback from these local experts were not obtained and shared, it was likely that the original cat–cow exercise may not have been properly performed or accepted by the youth athletes when implemented, demonstrating the value in consulting and working with experts. The inclusion of youth athlete end-users in the development process would also be advantageous; however, it would be purposeful to recognize that youth athletes are only at the beginning of their athletic career and are likely to have limited injury awareness, knowledge, and experience [34]. As such, there should be a limit to the extent of their involvement in the development process. For instance, in the development of YoSO-IPP, youth athletes were consulted on the feasibility of the exercise program after the program achieved consensus among the content experts (i.e., sports physicians and sports physiotherapists) and context experts (i.e., coaches) [33]. A feedback session was conducted where they were encouraged to share their views and opinions or raise any questions pertaining to any of the exercises, which led to a change in stretching duration from 30 s to 20 s for the cross-body stretch [33]. This approach ensured that youth athlete end-user feedback could still be considered despite their limited experience, to allow for further fine-tuning of the program where necessary.

It is to be emphasized that as the YoSO-IPP was developed specifically for overhead youth athletes in Singapore, with input from local experts and tailored to the unique constraints of the Singapore context, and that the direct translation of YoSO-IPP to any other context may not be as effective. Modifications to YoSO-IPP may be necessary to ensure its relevance to any other context (e.g., different age group of athletes, different country) by seeking feedback from the experts (and end-users) to adapt the program where necessary.

## 5. Step 4: Context-Specific Program Implementation and Evaluation

Implementation of the context-specific injury prevention program to assess its effects is the last step of the injury prevention process. A priority consideration at this step is to prevent or reduce bias during the implementation process following an understanding of the context. For instance, a key consideration in youth sport injury prevention program implementation is the possible bias introduced when coaches are tasked to deliver the program to their youth athletes. Youth athletes are in a dependent relationship with the coaches [35]. As influential agents in proximity of youth athletes, the motivation of youth sport coaches towards sport injury prevention is likely to influence the youth athletes’ willingness to participate or comply with the implemented program [36,37]. Therefore, it would be prudent for the research team to take on the role of delivery agents in the study and work with the youth athletes directly, to reduce or even eliminate coaches’ influence on athletes’ willingness to participate and their compliance with the program. While coaches may assist in coordinating the research schedule with the competition or training schedule, it would be methodologically and ethically appropriate to limit their involvement with program delivery or program implementation to reduce bias related to possible coercion. This would contribute to an objective evaluation of the effects of the program.

Another critical consideration is the increasing discussion for effective, yet time-efficient exercise-based injury prevention programs, which would contribute to the maximization of typically limited resources (e.g., time, equipment, personnel) in youth sports [36,38,39]. This has led to a shift in focus from outcome measures of injury incidence and severity (injury burden) to performance-related outcome measures [40,41,42]. Some examples of performance-related measurements include ball velocity, shoulder external rotation eccentric strength (i.e., strength of the shoulder external rotators when performing eccentric contraction), and shoulder internal rotation range of motion (i.e., maximum internal rotation range of the shoulder). The precise evaluation of performance outcome measures associated with the prescribed exercises would provide researchers with the ability to distinguish effective exercises from the non-essential exercises. The identification of effective exercises that likely accounted for the observed reduced injury risk through improved performance-related outcomes (e.g., improved shoulder rotation strength) would contribute to further refinement of the program for improved efficiency. Additionally, with performance-related outcome measures, the exercise-based injury prevention programs exhibit their additional capability of enhancing performance apart from reducing injury risk, which would potentially promote uptake and reduce attrition [37]. Therefore, the use of performance-related outcome measures would be desirable to allow for further program refinement, if necessary, while clearly demonstrating the ability of the exercises to improve performance at this step. The detailed process of this step of the OH-YA overuse injury prevention program is presented in Figure 5.

As with all intervention studies, it would be ideal to conduct the program and evaluate its effects in a longitudinal study. This could provide insights into any dose–response relationship that the program may have, including a ceiling effect where effects may plateau. However, conducting a longitudinal study may be challenging due to the limited resources commonly observed in youth team sports. Moreover, youth sport seasons are typically shorter and in the case of Singapore, mostly last 2–3 months in the calendar year, which may be a constraint for longitudinal studies. A longitudinal study may also be less appealing to the coaches, who may consider the intervention as a disruption to their training plans. Therefore, it is necessary to communicate and work with the coaches to understand their beliefs and attitudes toward injury prevention, including the equipment available, before implementing the developed program in any youth sport context to minimize challenges.

## 6. Conclusions

The OverHead Youth Athlete (OH-YA) overuse injury prevention process presented provides an enhanced injury prevention model with a comprehensive illustration of the importance of context, including how it can be integrated into the injury prevention process among youth overhead athletes to minimize the “research-to-practice” gap at every step of the process and contribute to the maximization of limited resources. However, it must be noted that on a single-unit level, applying the OH-YA overuse injury prevention process uniformly across all youth overhead sport settings without adjustments may not be appropriate as each sporting context still has its own unique factors (e.g., time constraints in the team’s practice schedule, qualifications of personnel, stakeholder support). Therefore, future research may seek to refine the framework to better suit the specific needs of their context to further validate the OH-YA injury prevention process. This may also lead to other future research of applying the OH-YA model to other types of youth injuries, to explore its potential in addressing a broader range of injury risks. A thorough understanding of context in any injury prevention efforts cannot be understated and is vital in supporting the development of an inclusive intervention for sport injury prevention and management.

## Figures and Tables

**Figure 1 jcm-14-00971-f001:**
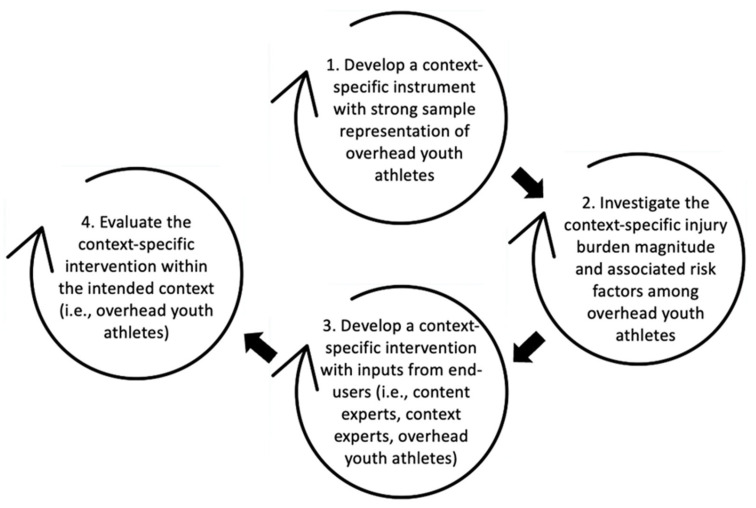
The OverHead Youth Athlete (OH-YA) overuse injury prevention process.

**Figure 2 jcm-14-00971-f002:**
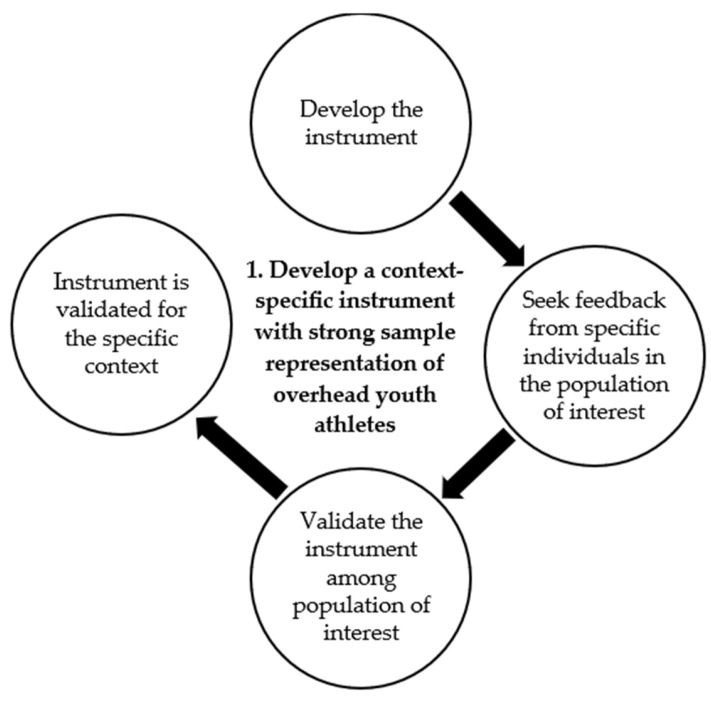
Detailed illustration of Step 1 of the OverHead Youth Athlete (OH-YA) overuse injury prevention process.

**Figure 3 jcm-14-00971-f003:**
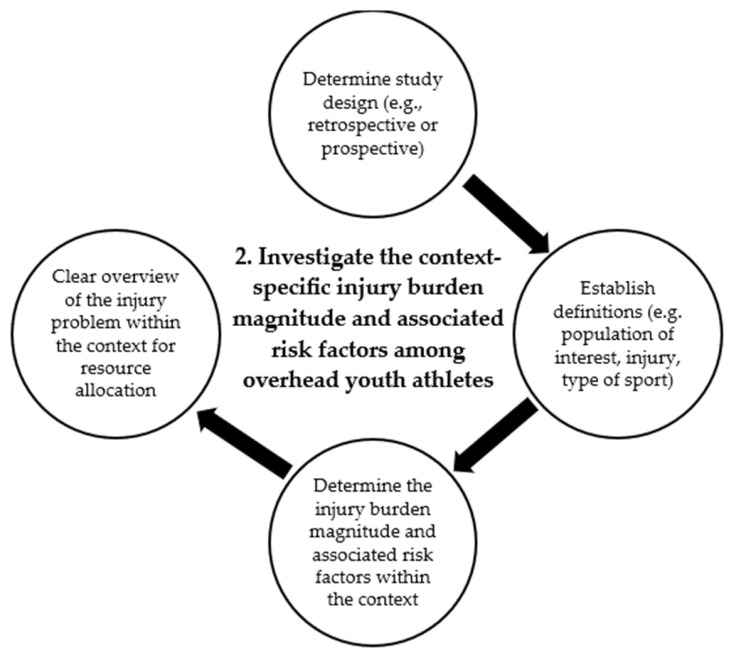
Detailed illustration of Step 2 of the OverHead Youth Athlete (OH-YA) overuse injury prevention process.

**Figure 4 jcm-14-00971-f004:**
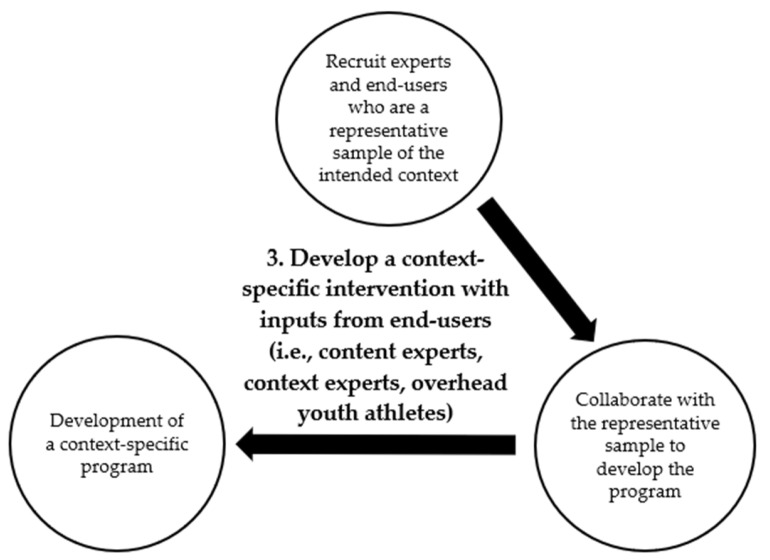
Detailed illustration of Step 3 of OverHead Youth Athlete (OH-YA) overuse injury prevention process.

**Figure 5 jcm-14-00971-f005:**
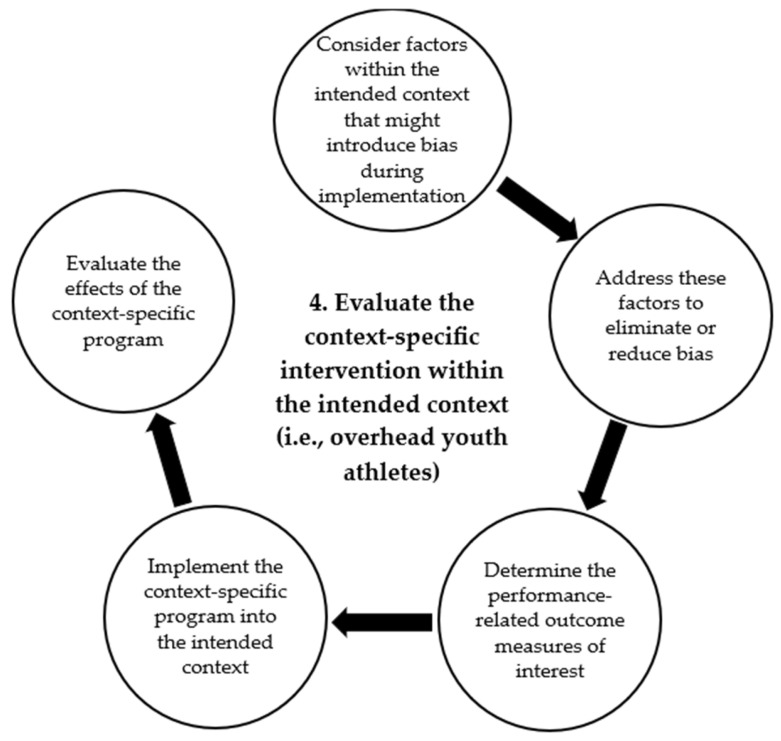
Detailed illustration of Step 4 of the OverHead Youth Athlete (OH-YA) overuse injury prevention process.

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
