# Peer review of "Moving Beyond the Checklist—An Enhanced Approach to Context-Driven Overuse Injury Prevention in Overhead Youth Athletes"

_jcm, 2025, doi:10.3390/jcm14030971_

Round 1
Reviewer 1 Report
Comments and Suggestions for Authors
I would like to commend the authors for their well-executed work on this manuscript. The proposed framework for context-driven overuse injury prevention in overhead youth athletes is innovative and highly relevant.
Below, I provide specific suggestions to further enhance the quality and impact of the review.
Abstract:
The abstract is clear and summarizes the article effectively. However, the introduction of the OH-YA process could be made more compelling by explicitly stating how it improves upon existing methods. Consider including a brief mention of the anticipated impact or novelty of the framework.
Introduction:
1) The introduction provides a solid foundation and justification for a context-driven approach. However, it could benefit from:
- A clearer delineation between the gaps in existing models and how the OH-YA framework addresses them.
- Explicitly stating how the review advances the "context-driven sequence of prevention" model (Bolling et al., 2018).
Step 1
The detailed description of the development and validation of the Youth Overuse Injury Questionnaire (YOvIQ) is excellent. To enhance clarity:
- Provide a brief explanation of how YOvIQ differs from or improves upon the Oslo Sport Trauma Research Center Overuse Injury Questionnaire (OSTRC-O2) for this population.
- Consider adding more specifics about the feedback process from end-users and content experts.
Step 2
This section effectively highlights the importance of defining the injury burden magnitude and associated factors. However:
- Include more detail on how the injury categories (severe, moderate, mild) are operationalized and their practical implications.
- Discuss how findings from studies like Lau & Mukherjee (2023) might generalize to other populations or contexts.
Step 3
The emphasis on collaboration and ecological validity in program development is commendable. To strengthen this section:
- Provide examples of how feedback from youth athletes and coaches led to specific adjustments in the YoSO-IPP.
Conclusions
The conclusion reiterates the article's main points but could be more forward-looking. Consider:
- Highlighting potential challenges in adopting the OH-YA process in diverse settings.
- Suggesting areas for future research to further validate and refine the framework.
Author Response
Response to Reviewer 1 Comments
Manuscript ID: jcm-3382578
Manuscript Title: Moving beyond the checklist – An enhanced approach to context-driven overuse injury prevention in overhead youth athletes
General comments:
Thank you very much for taking the time to review this manuscript. Please find the detailed responses below and the corresponding revisions/corrections highlighted/in track changes in the re-submitted files.
Reviewer’s comments
I would like to commend the authors for their well-executed work on this manuscript. The proposed framework for context-driven overuse injury prevention in overhead youth athletes is innovative and highly relevant.
Below, I provide specific suggestions to further enhance the quality and impact of the review.
Response: We thank the reviewer for the kind comments. Please find our responses and documented revisions below.
Abstract:
The abstract is clear and summarizes the article effectively. However, the introduction of the OH-YA process could be made more compelling by explicitly stating how it improves upon existing methods. Consider including a brief mention of the anticipated impact or novelty of the framework.
Response: Thank you for the suggestion. We have added in the sentence “This framework would likely help maximize the use of typically limited resources in youth sports.” in the abstract as a brief mention of the anticipated impact of the framework.
Introduction:
1) The introduction provides a solid foundation and justification for a context-driven approach. However, it could benefit from:
- A clearer delineation between the gaps in existing models and how the OH-YA framework addresses them.
- Explicitly stating how the review advances the "context-driven sequence of prevention" model (Bolling et al., 2018).
Response: Thank you the comment. A sentence has been included to highlight the gap in the literature and how the OH-YA framework contributes to addressing the gap – “Despite these numerous concerns, seemingly no injury prevention process framework currently exists to address shoulder overuse inures within the specific context of overhead youth athletes.”. This sentence comes before the existing sentences, “Therefore, we propose a specific approach for overuse injury prevention in overhead youth athletes, while explicitly integrating and emphasizing the importance of context within the injury prevention process. An enhanced, sequenced and inclusive model for overuse injury prevention in overhead youth athletes, known as the OverHead Youth Athlete (OH-YA) overuse injury prevention process…” to highlight the gap in the literature and how the OH-YA framework addresses this gap.
How this review advances the “context-driven sequence of prevention" model (Bolling et al., 2018)” have been included at the end of the Introduction section – “The OH-YA overuse injury prevention process builds on the “context-driven sequence of prevention” by introducing a preliminary step of instrument validation to enhance data accuracy. The framework will also likely help maximize the use of limited resources in youth sports.”. Thank you.
Step 1
The detailed description of the development and validation of the Youth Overuse Injury Questionnaire (YOvIQ) is excellent. To enhance clarity:
- Provide a brief explanation of how YOvIQ differs from or improves upon the Oslo Sport Trauma Research Center Overuse Injury Questionnaire (OSTRC-O2) for this population.
- Consider adding more specifics about the feedback process from end-users and content experts.
Response: Thank you for the comment.
A brief explanation of how YOvIQ differs from OSTRC-O2 have been included – “YOvIQ was developed by building on the established updated Oslo Sport Trauma Research Center Overuse Injury Questionnaire (OSTRC-O2) to make it context-specific to youth athletes - this included adjusting the recall period, reorganizing the question order, emphasizing key words by bolding and underlining them, and adding a short, open-ended question for injury description (Lau & Mukherjee, 2024b).”
Some specifics of the feedback process have been included – “The initial draft of YOvIQ was shared with the context experts who provided feedback via email. Subsequently, an online feedback session was conducted with the youth athlete end-users on the second draft of YOvIQ, to enhance the suitability of the questionnaire. Notably, it should be highlighted that when seeking feedback from youth athletes, it is important to emphasize the anonymity of their feedback, to help create a safe environment where they feel comfortable to share their opinions freely (Lau & Mukherjee, 2024b).”
Step 2
This section effectively highlights the importance of defining the injury burden magnitude and associated factors. However:
- Include more detail on how the injury categories (severe, moderate, mild) are operationalized and their practical implications.
- Discuss how findings from studies like Lau & Mukherjee (2023) might generalize to other populations or contexts.
Response: Thank you for the comment.
More details on the implications of injury categories have been included - “Having these categorizations is essential as it provides a quick overview of injury levels within the context of interest (e.g., a sports team), which can help prioritize prevalent or pressing injury concerns to allow for more efficient allocation of limited resources. Additionally, the strongly representative and valid data collected may also be further analyzed to identify factors associated with the injury within the context (e.g., age, level of sport participation, gender), which will assist coaches and/or athletic trainers to promptly interpret and refine their injury management approaches.”
A discussion on the generalization of findings from studies like Lau & Mukherjee (2023) have been included – “Considering the relatively large sample size of 434 participants across 11 overhead sports, the findings from this study, and similar studies alike, can provide valuable insights into injury patterns and risk factors for athletes in these sports. These insights may also contribute to the development of more targeted injury prevention interventions to foster safer sport participation while enhancing performance.”
Step 3
The emphasis on collaboration and ecological validity in program development is commendable. To strengthen this section:
- Provide examples of how feedback from youth athletes and coaches led to specific adjustments in the YoSO-IPP.
Response: Thank you for the comment.
The example for how feedback from coaches led to specific adjustments in the YoSO-IPP is now included – “The importance of seeking feedback from experts was demonstrated in our development of the Singapore Youth Shoulder Overuse Injury Prevention Programme (YoSO-IPP) where we consulted local experts via a two-round Delphi technique (Lau & Mukherjee, 2024a). In our study, as part of the iterative multistage process, the experts utilized a 5-point Likert Scale to provide feedback and were encouraged to leave comments for anonymous sharing with the other experts in the next round. To illustrate an example, some experts left comments in the first round of the Delphi technique regarding the potential difficulty of implementing the cat-cow exercise among the targeted participants of 12-18 year-old athletes, due to its technical difficulty. This led the research team to modify the cat-cow exercise to make it easier to perform, which was included in the second round and subsequently reached consensus (i.e., 75% agreement) among the experts (Lau & Mukherjee, 2024a). If feedback from these local experts were not obtained and shared, it was likely that the original cat-cow exercise may not have been properly performed or accepted by the youth athletes when implemented, demonstrating the value in consulting and working with experts.”.
The example for how feedback from youth athletes led to specific adjustments in the YoSO-IPP is now included – “A feedback session was conducted where they were encouraged to share their views, opinions or raise any questions pertaining to any of the exercises, which led to a change in stretching duration from 30 seconds to 20 seconds for the cross-body stretch (Lau & Mukherjee, 2024a).”
Conclusions
The conclusion reiterates the article's main points but could be more forward-looking. Consider:
- Highlighting potential challenges in adopting the OH-YA process in diverse settings.
- Suggesting areas for future research to further validate and refine the framework.
Response: Thank you for the comment.
The potential challenges in adopting the OH-YA process in diverse settings, and correspondingly, suggested future research has been included – “However, it must be noted that on a single-unit level, applying the OH-YA overuse injury prevention process uniformly across all youth overhead sport settings without adjustments may not be appropriate as each sporting context still has its own unique factors (e.g., time constraints in the team’s practice schedule, qualifications of personnel, stakeholder support). Therefore, future research may seek to refine the framework to better suit the specific needs of their context to further validate the OH-YA injury prevention process. This may also lead to other future research of applying the OH-YA model to other types of youth injuries, to explore its potential in addressing a broader range of injury risks.”.
Reviewer 2 Report
Comments and Suggestions for Authors
1. Introduction
Comment:
The well-written introduction highlights the necessity of a context-specific framework for preventing injuries in young athletes. It is commendable that this study addresses the differences between traditional and context-based approaches.
Suggestion:
Elaborate terms such as "research-to-practice gap" and "context-driven sequence of prevention" to make the section more accessible to a broader audience.
Specific examples of athletes or scenarios in which these differences had a significant impact should be included.
2. Methodology
Comment:
The proposed methodology is well-defined, including steps such as developing and validating context-specific instruments. It is noteworthy that experts and end-users were involved to ensure relevance.
Suggestion:
Experts and athletes should be provided with more details regarding the feedback process, including the challenges encountered and how they were addressed.
Discuss why a specific instrument, such as the Youth Overuse Injury Questionnaire (YOvIQ), is more effective than other existing tools.
3. Determining the magnitude of injuries
Comment:
The study's use of a retrospective design and the establishment of clear inclusion criteria are well structured. The presentation of specific risk factors and injury severity categories is useful.
Suggestion:
Explain in more detail how definitions for "overuse injuries" and "sporting context" were established.
Add a brief section on how the obtained results can be generalized to other athletic categories or disciplines.
4. Development of a context-specific intervention
Comment:
Involving end-users and experts in program development is well documented. The Singapore Youth Shoulder Overuse Injury Prevention Programme (YoSO-IPP) is a highly relevant example.
Suggestion:
More details regarding the Delphi technique used in developing the YoSO-IPP program should be included. How did it facilitate consensus among experts?
Present possible improvements for adapting the intervention to other countries or sports.
5. Implementation and evaluation
Comment:
This study correctly emphasizes the importance of minimizing bias during implementation. The use of performance-based measurements is an innovative and welcome approach.
Suggestion:
Provide detailed examples of performance-based measurements used (e.g., shoulder rotation and strength).
Discuss the duration of the program’s effects evaluation: Is short-term evaluation sufficient, or should it extend to long-term assessments?
To address the challenges faced during implementation, such as resistance from coaches or limited resources.
6. Conclusions
Comment:
The conclusions effectively summarize the importance of the context-based approach and its potential impact.
Suggestion:
Include a section with practical recommendations for implementing the OH-YA model at the local or international level.
Further research opportunities should be highlighted, such as extending the model to prevent other types of injuries.
7. References
Comment:
The list of references is extensive and well-updated, including relevant and reliable sources.
Suggestion:
Check if there are recent works from 2023 to 2024 that could be added to emphasize the latest advancements.
Consider creating an additional category in the bibliography for key sources defining the "context-driven sequence of prevention" model.
Author Response
Response to Reviewer 2 Comments
Manuscript ID: jcm-3382578
Manuscript Title: Moving beyond the checklist – An enhanced approach to context-driven overuse injury prevention in overhead youth athletes
General comments:
Thank you very much for taking the time to review this manuscript. Please find the detailed responses below and the corresponding revisions/corrections highlighted/in track changes in the re-submitted files.
- Introduction
Comment:
The well-written introduction highlights the necessity of a context-specific framework for preventing injuries in young athletes. It is commendable that this study addresses the differences between traditional and context-based approaches.
Suggestion:
Elaborate terms such as "research-to-practice gap" and "context-driven sequence of prevention" to make the section more accessible to a broader audience.
Response: Thank you for the suggestion. We have elaborated on the terms.
(1) Research-to-practice gap : “The significance of context in sport injury prevention has recently received growing attention in parallel with focused efforts to address the challenge of translating research findings into real-world conditions, otherwise known as the research-to-practice gap (Bolling et al., 2018; Verhagen, 2012).”
(2) Context-driven sequence of prevention: “Essentially, the model calls for an all-encompassing focus on the intended context to drive the injury prevention process, with the goal of enhancing the application of injury prevention research in real-world settings.”
Specific examples of athletes or scenarios in which these differences had a significant impact should be included.
Response: Thank you for the suggestion. We have included this example in the introduction – “For instance, it was documented in a qualitative study among junior high school teachers and students that the implemented injury prevention program (i.e., iSPRINT) was not time-efficient as participants had to spend five to eight minutes setting up the necessary equipment, and a similar amount of time to keep them afterwards (Richmond et al., 2020). This led them to make their own modifications to the program (i.e., execute the balance-related exercises without a wobble board), which may have affected the intended effects of the program (Richmond et al., 2020).”
- Methodology
Comment:
The proposed methodology is well-defined, including steps such as developing and validating context-specific instruments. It is noteworthy that experts and end-users were involved to ensure relevance.
Suggestion:
Experts and athletes should be provided with more details regarding the feedback process, including the challenges encountered and how they were addressed.
Response: Thank you for the suggestion. We have included this as such – “The initial draft of YOvIQ was shared with the context experts who provided feedback via email. Subsequently, an online feedback session was conducted with the youth athlete end-users on the second draft of YOvIQ, to enhance the suitability of the questionnaire. Notably, it should be highlighted that when seeking feedback from youth athletes, it is important to emphasize the anonymity of their feedback, to help create a safe environment where they feel comfortable to share their opinions freely (Lau & Mukherjee, 2024b).”
Discuss why a specific instrument, such as the Youth Overuse Injury Questionnaire (YOvIQ), is more effective than other existing tools.
Response: Thank you for the suggestion. We wish to make a point here that the reader can always refer to our paper to have more information on how the YOvIQ fares against the other instrument (OSTRC-O2)–
Lau & Mukherjee (2024). Development and validation of an overuse injury questionnaire for youth athletes: The Youth Overuse Injury Questionnaire. Physical Therapy in Sport, 67, 47-53. https://doi.org/10.1016/j.ptsp.2024.03.003.
However, we agree with the Reviewer on elaborating more on the effectiveness on YOvIQ as a youth athlete centered instrument for documenting overuse injuries. To further emphasize the strengths/advantages of YOvIQ, we have included these short paragraphs within the section of “Step 1 Ensure that the instrument used to determine the injury magnitude burden is valid for the context”:
(1) “YOvIQ was developed by building on the established updated Oslo Sport Trauma Research Center Overuse Injury Questionnaire (OSTRC-O2) to make it context-specific to youth athletes - this included adjusting the recall period, reorganizing the question order, emphasizing key words by bolding and underlining them, and adding a short, open-ended question for injury description (Lau & Mukherjee, 2024b).”
(2) “By specifically tailoring the questionnaire to make it suitable for use among competitive youth athletes, seeking expert and end-user feedback, and further validating its use among youth athletes from a diverse range of sports, YOvIQ can be regarded as a highly validated instrument within the specific context of competitive youth athletes. This comprehensive validation process of YOvIQ also offers enhanced accuracy of the data collected, enabling better tailored context-specific preventive interventions for increased effectiveness.”
- Determining the magnitude of injuries
Comment:
The study's use of a retrospective design and the establishment of clear inclusion criteria are well structured. The presentation of specific risk factors and injury severity categories is useful.
Suggestion:
Explain in more detail how definitions for "overuse injuries" and "sporting context" were established.
Response: Thank you for the suggestion.
The definition for overuse injury is now included – “The presence of an overuse injury was registered if participants reported anything less than full participation ability in training or competitions.”
As we are unable to locate the term “sporting context” within this section, we defined the term “overhead sport” included in the inclusion criteria– “(2) the primary sport participated in is an overhead sport where the arm is repetitively raised above the head to strike or throw an object,…”
Add a brief section on how the obtained results can be generalized to other athletic categories or disciplines.
Response: Thank you for the suggestion. We have addressed this here – “Considering the relatively large sample size of 434 participants across 11 overhead sports, the findings from this study, and similar studies alike, can provide valuable insights into injury patterns and risk factors for athletes in these sports. These insights may also contribute to the development of more targeted injury prevention interventions to foster safer sport participation while enhancing performance.”
- Development of a context-specific intervention
Comment:
Involving end-users and experts in program development is well documented. The Singapore Youth Shoulder Overuse Injury Prevention Programme (YoSO-IPP) is a highly relevant example.
Suggestion:
More details regarding the Delphi technique used in developing the YoSO-IPP program should be included. How did it facilitate consensus among experts?
Response: Thank you for the suggestion. We have included a paragraph to share more about the Delphi Technique – “The importance of seeking feedback from experts was demonstrated in our development of the Singapore Youth Shoulder Overuse Injury Prevention Programme (YoSO-IPP) where we consulted local experts via a two-round Delphi technique (Lau & Mukherjee, 2024a). In our study, as part of the iterative multistage process, the experts utilized a 5-point Likert Scale to provide feedback and were encouraged to leave comments for anonymous sharing with the other experts in the next round. To illustrate an example, some experts left comments in the first round of the Delphi technique regarding the potential difficulty of implementing the cat-cow exercise among the targeted participants of 12-18 year-old athletes, due to its technical difficulty. This led the research team to modify the cat-cow exercise to make it easier to perform, which was included in the second round and subsequently reached consensus (i.e., 75% agreement) among the experts (Lau & Mukherjee, 2024a). If feedback from these local experts were not obtained and shared, it was likely that the original cat-cow exercise may not have been properly performed or accepted by the youth athletes when implemented, demonstrating the value in consulting and working with experts.”
Present possible improvements for adapting the intervention to other countries or sports.
Response: Thank you for the suggestion. We have included this paragraph to talk about program adaption in other contexts – “It is to be emphasised that as the YoSO-IPP was developed specifically for youth athletes in Singapore, with input from local experts and tailored to the unique constraints of the Singapore context, and that the direct translation of YoSO-IPP to any other context may not be as effective. Modifications to YoSO-IPP may be necessary to ensure its relevance to any other context (e.g., different sport, different age group of athletes, different country) by seeking feedback from the experts (and end-users) to adapt the program where necessary.”
- Implementation and evaluation
Comment:
This study correctly emphasizes the importance of minimizing bias during implementation. The use of performance-based measurements is an innovative and welcome approach.
Suggestion:
Provide detailed examples of performance-based measurements used (e.g., shoulder rotation and strength).
Response: Thank you for the suggestion. This has been included – “Some examples of performance-based measurements include ball velocity, shoulder external rotation eccentric strength (i.e., strength of the shoulder external rotators when performing eccentric contraction), and shoulder internal rotation range of motion (i.e., maximum internal rotation range of the shoulder).”
Discuss the duration of the program’s effects evaluation: Is short-term evaluation sufficient, or should it extend to long-term assessments?
To address the challenges faced during implementation, such as resistance from coaches or limited resources.
Response: Thank you for the suggestions. We have decided to address them together –
“As with all intervention studies, it would be ideal to conduct the program and evaluate its effects in a longitudinal study. This could provide insights into any dose-response relationship that the program may have, including a ceiling effect where effects may plateau. However, conducting a longitudinal study may be challenging due to the limited resources commonly observed in youth team sports. Moreover, youth sport seasons are typically shorter and mostly last 2-3 months in the calendar year, which may be a constraint for longitudinal studies. A longitudinal study may also be less appealing to the coaches, who may consider the intervention as a disruption to their training plans. Therefore, it is necessary to communicate and work with the coaches to understand their belies and attitudes to injury prevention, including the equipment available, before implementing the developed program in any youth sport context to minimize challenges.”
- Conclusions
Comment:
The conclusions effectively summarize the importance of the context-based approach and its potential impact.
Suggestion:
Include a section with practical recommendations for implementing the OH-YA model at the local or international level.
Further research opportunities should be highlighted, such as extending the model to prevent other types of injuries.
Response: Thank you for the suggestions. We have decided to address them together – “However, it must be noted that on a single-unit level, applying the OH-YA overuse injury prevention process uniformly across all youth overhead sport settings without adjustments may not be appropriate as each sporting context still has its own unique factors (e.g., time constraints in the team’s practice schedule, qualifications of personnel, stakeholder support). Therefore, future research may seek to refine the framework to better suit the specific needs of their context to further validate the OH-YA injury prevention process. This may also lead to other future research of applying the OH-YA model to other types of youth injuries, to explore its potential in addressing a broader range of injury risks.”
- References
Comment:
The list of references is extensive and well-updated, including relevant and reliable sources.
Suggestion:
Check if there are recent works from 2023 to 2024 that could be added to emphasize the latest advancements.
Response: Thank you for the suggestion. We have included two more references, from 2022 and 2024, which can be found as citations for this sentence:
“This can be achieved through collaboration with a strongly representative sample of the context during program development (Ageberg et al., 2022; Donaldson et al., 2016; Mann et al., 2024).”
Ageberg, E., Brodin, E. M., Linnéll, J., Moesch, K., Donaldson, A., Adébo, E., Benjaminse, A., Ekengren, J., Granér, S., Johnson, U., Lucander, K., Myklebust, G., Møller, M., Tranaeus, U., & Bunke, S. (2022). Cocreating injury prevention training for youth team handball: bridging theory and practice. BMJ Open Sport & Exercise Medicine, 8(2), e001263. https://doi.org/10.1136/bmjsem-2021-001263
Mann, R. H., Clift, B. C., Day, J., & Barker, A. R. (2024). Co-creation of injury prevention measures for competitive adolescent distance runners: knowledge, behavior, and needs of athletes and coaches enrolled on England Athletics' Youth Talent Programme. Annals of Medicine, 56(1), 2334907. https://doi.org/10.1080/07853890.2024.2334907
Consider creating an additional category in the bibliography for key sources defining the "context-driven sequence of prevention" model.
Response: Thank you for the suggestion. We have decided not to proceed with this as the context-driven sequence or prevention is defined/from one paper only – Bolling, C., van Mechelen, W., Pasman, H. R., & Verhagen, E. (2018). Context matters: Revisiting the first step of the ‘sequence of prevention’ of sports injuries. Sports Medicine, 48(10), 2227-2234. https://doi.org/10.1007/s40279-018-0953-x